# Immunometabolic Profiling of Chronic Subdural Hematoma through Untargeted Mass Spectrometry Analysis: Preliminary Findings of a Novel Approach

**DOI:** 10.3390/diagnostics13213345

**Published:** 2023-10-30

**Authors:** Maria Kipele, Michael Buchfelder, R. Verena Taudte, Andreas Stadlbauer, Thomas Kinfe, Yavor Bozhkov

**Affiliations:** 1Department of Neurosurgery, Erlangen University, 91054 Erlangen, Germany; maria.kipele@uk-erlangen.de (M.K.); michael.buchfelder@uk-erlangen.de (M.B.); andreas.stadlbauer@uk-erlangen.de (A.S.); 2Core Facility for Metabolomics, Department of Medicine, Philipps University Marburg, 35037 Marburg, Germany; verena.taudte@uni-marburg.de; 3Division of Stereotactic and Functional Neurosurgery, Department of Neurosurgery, Erlangen University, 91054 Erlangen, Germany; thomasmehari.kinfe@uk-erlangen.de

**Keywords:** metabolomics, chronic subdural hematoma, metabolic profile, inflammation

## Abstract

**Objective**: Metabolomics has growing importance in the research of inflammatory processes. Chronic subdural hematoma (cSDH) is considered to be, at least in part, of inflammatory nature, but no metabolic analyses yet exist. Therefore, a mass spectrometry untargeted metabolic analysis was performed on hematoma samples from patients with cSDH. **Methods**: A prospective analytical cross-sectional study on the efficacy of subperiosteal drains in cSDH was performed. Newly diagnosed patients had the option of granting permission for the collection of a hematoma sample upon its removal. The samples were analyzed using liquid chromatography–mass spectrometry to obtain different types of metabolites from diverse biochemical classes. The statistical analysis included data cleaning, imputation, and log transformation, followed by PCA, PLS-DA, HCA, and ANOVA. The postoperative course of the disease was followed for 3 months. The metabolite concentrations in the hematoma fluid were compared based on whether a recurrence of the disease was recorded within this time frame. **Results**: Fifty-nine samples from patients who were operated on because of a cSDH were gathered. Among those, 8 samples were eliminated because of missing metabolites, and only 51 samples were analyzed further. Additionally, 39 samples were from patients who showed no recurrence over the course of a 3-month follow-up, and 12 samples were from a group with later recurrence. We recorded a noticeable drop (35%) in the concentration of acylcarnitines in the ”recurrence group“, where 10 of the 22 tested metabolites showed a significant reduction (*p* < 0.05). Furthermore, a noticeable reduction in different Acyl-CoA-dehydrogenases was detected (VLCAD-deficiency *p* < 0.05, MCAD-deficiency *p* = 0.07). No further changes were detected between both populations. **Conclusions**: The current study presents a new approach to the research of cSDH. The measurements presented us with new data, which, to date, are without any reference values. Therefore, it is difficult to interpret the information, and our conclusions should be considered to be only speculative. The results do, however, point in the direction of impaired fatty acid oxidation for cases with later recurrence. As fatty acid oxidation plays an important role in inflammatory energy metabolism, the results suggest that inflammatory processes could be aggravated in cases with recurrence. Because our findings are neither proven through further analyses nor offer an obvious therapy option, their implications would not change everyday practice in the management of cSDH. They do, however, present a further possibility of research that might, in the future, be relevant to the therapy.

## 1. Introduction

The cSDH was first described nearly four centuries ago by Johann Wepfer [1]. At first, however, it was falsely viewed as a stroke-like lesion [1]. A century later, in 1751, James Hill described it as a “Black liquid blood” which appeared from under the dura. Houssard described the concept of membranes surrounding bleeding or clotting in 1817 and Bayle in 1826. Bayle even went further, suggesting that the lamination could be the result of recurrent bleeding [2]. At a later stage, “Pachymeningitis hemorrhagica chronica interna” was a term used by Virchow to describe cSDH. He indicated that trauma could be responsible for the initiation of cSDH, but the lesion itself was a product of chronic inflammation of the dura [3]. Nevertheless, at the beginning of the 20th century, the predominant theory based on publications by Trotter and Putnam was that cSDH was a traumatic disease [4]. Later, Markwalder was one of the scientists to postulate that, although of traumatic origin, subdural hematomas might be somewhat self-perpetuating in nature and associated with a complex underlying inflammatory mechanism [5]. Many theories about cSDH pathophysiology have been proposed over the years. Many have suggested a more complex intertwined pathway of microbleeds, inflammation, angiogenesis, and local coagulopathy as being involved [6,7]. These inflammatory processes and their contributions to the development and maintenance of cSDH are now more clearly understood. Nevertheless, it should be noted that the current theories only represent the current state of knowledge and might be subject to revision in the future.

As a consequence of mostly fairly minor head trauma, a cleavage of the dural border cell layer develops. The tearing of the dural border cell layer causes the bleeding, rebleeding, and extravasation of the CSF in subdural space. This, in turn, leads to an accumulation of fibroblasts and the formation of a hematoma membrane [8]. The membrane neo-vessels, through their abnormal permeability, allow for the extravasation of inflammatory mediators into the hematoma cavity. Masatoshi Kitazono reported the involvement of inflammatory cytokines in hematoma formation. This includes thrombomodulin, as well as IL-6, IL-10, and TNF alpha [9]. Fibroblast growth signals, when activated, cause further inflammatory and angiogenic responses that form a well-vascularized external neo-membrane. Neo-membranes are rich in small blood vessels which are kept leaky through MEK/ERK, and these transducers are activated by PIGF and VEGF, which are found in neo-membranes; IL 10 and 13 are also reported to contribute to the chronification of cSDH [10]. VEGF is the most potent stimulator of permeability currently known and is reported to increase blood flow, microvascular permeability, and angiogenesis [11]. VEGF involvement in the formation of cSDH is described by its higher concentration in hematomas than in serum [12,13]. PIGF and sVEGFR-1 were also reported to be relatively higher in hematomas than in the sera of the patients with cSDH, which showed PIGF involvement in local inflammation and angiogenesis [14]. Proteomes such as carbonic anhydrase I are reported to be involved in cSDH formation [15]. Higher levels of chemokine CCL2, CXCL8, CXCL 9, and CXCL10 in hematomas than in sera are also reported to be responsible for the cSDH formation [16].

The ensuing inflammation is a local process that does not affect the whole organism. This inflammation is likely perpetuated by the fibroblasts and other inflammatory cells, such as eosinophils, which line the membranes. Angiogenesis is induced by some of the inflammatory pathways and leads to the creation of immature capillaries and recurrent microbleeds. Because of induced hyperfibronolysis and coagulopathy, no clot formation is observed. This inflammatory cycle leads to the development and enlargement of the hematoma until the beginning of clinical symptoms leads to its diagnosis and treatment.

Because the inflammation is locally contained, the first symptoms of a cSDH appear because of the mass effect. Depending on its size, the first line of treatment is almost always direct decompression via a small skull opening. While this mechanically alleviates brain compression, a further mechanism of action might include the removal of a large part of the inflammatory components within the hematoma. Still, one of the most common issues of cSDH is frequent recurrence. This may be because further inflammatory processes can take place. As a means of reducing recurrent cases, further therapeutic strategies have been developed. These range from pharmacological agents such as dexamethasone and tranexamic acid to MMA embolization as a way to stop angiogenesis [8,17,18,19,20].

Nevertheless, despite recent progress in the understanding of the pathophysiology, there is still a need for further investigation into the topic as recurrence continues to pose a relevant problem in everyday practice. It is yet unclear whether the level of inflammation plays a role in the development of recurrence. Therefore, we decided to compare the inflammatory profiles of hematoma fluids between patients with and without later disease recurrence. We opted for a different method to what has been used across the literature; as we focused on metabolites, we used liquid chromatography–mass spectrometry to examine the selected probes. Metabolomics has, so far, not played an important part in cSDH research, with only a few authors investigating the role of metabolites through chromatography [21].

## 2. Materials and Methods

### Study Design

The current study was originally planned as a part of a separate prospective randomized trial regarding surgery of cSDH with adjuvant use of periosteal drains. Ethical clearance for this study was granted by the ethical committee of the University Clinic of Erlangen (440_19B).

The enrollment of 100 operative cases for newly diagnosed cSDH was originally planned (88 patients; 12 with bilateral cSDH were enrolled) (Table 1). Evacuation of the hematoma was performed via a single extended burr-hole craniostomy. Upon acquiring patients’ informed consent, patients were given the extra option of granting permission to the collection of a small sample of the cSDH fluid upon evacuation. Altogether, 59 samples of hematoma fluid (10 mL) were collected during these procedures. Patients were followed up for an overall period of 3 months, and the development of a recurrence was tracked. Recurrence was defined as a non-traumatic recollection of blood in the subdural space with the need for an operative revision. During the follow-up period, no antiplatelet medication or other anticoagulants were applied, although pain medication in most cases included NSARs.

The collected samples of hematoma were then each stored at a temperature of −80 °C in the laboratory, where an LC–MS analysis was performed on a Dionex Ultimate^TM^ 3000 chromatographic system coupled to a Q Exactive^TM^ Focus mass spectrometer (Thermo Fisher Scientific, Dreieich, Germany). The metabolite analysis was performed with a ready-to-use, standardized kit for broad lipid and metabolic profiling (AbsoluteIDQ^®^ p400 HR, Biocrates Life Sciences AG, Innsbruck, Austria) (Figure 1).

Almost 400 metabolites were quantified. Among those were immunometabolic and oxidative stress markers such as phosphatidylcholines, triglycerides, diglycerides, sphingomyelins, ceramides, cholesteryl esters, amino acids, biogenic amines, and acylcarnitines (Figure 2).

After the planned 3-month observational period, 46 samples were categorized as samples from patients without recurrence, and 13 were categorized as samples from patients later suffering from recurrent cSDH. Within the measurement of these 59 samples, it was apparent that some of them presented with missing values; therefore, 8 samples were completely removed from the dataset (Table 2). Imputation was performed on these 51 remaining samples following a K-nearest neighbor approach.

To exclude the metabolites with too many concentration values below the limit of detection (LOD), a general cleaning of data was performed. Among 400 metabolites, only 238 metabolites remained. The cleaned and imputed dataset was then used for data transformation and statistical analysis. Most of the removed metabolites were acylcarnitines, triglycerides, and phosphatidylcholines. The dataset was further processed via either Log2- or Log10-transformation. The METIDQ metaboINDIKATOR tool was used to calculate the sums and ratios of the cleaned and imputed data.

## 3. Definition of Objectives

The primary aim of this study was to acquire a metabolic phenotype of the cSDH fluid. Furthermore, it was to be investigated whether hematomas presenting with later recurrence showed any difference compared to those with a more benign course of the disease. Recurrence was defined as the need for operative revision due to neurological symptoms or radiological progression. The results of both groups were compared.

### Statistical Analysis

Various statistical methods were used to identify differences in metabolite levels between the statistical groups. Liquid chromatography–mass spectrometry was initially used to analyze the samples. The statistical analysis included data cleaning (descriptive statistics), imputation (univariate statistics), and log transformation (multivariate statistics). In this study, principal component analysis (PCA), partial least squares-discriminant analysis (PLS-DA), and hierarchical cluster analysis (HCA) were used as multivariate approaches and analysis of variance (ANOVA).

General measures of central tendency and dispersion were calculated for the cleaned and imputed dataset to provide a quantitative description of the study groups. Prior to statistical group comparison analysis, log10-transformation was conducted on the imputed data. Statistical analysis was then performed via a fixed linear model analysis of variance (ANOVA) for the differences in metabolite concentrations between groups. The significance level was set to 0.05, and *p*-values were calculated.

All statistical analyses were performed by Biocrates. MetIDQ, Biocrates’ in-house software, Version V01-2020, was used for raw data analysis, data evaluation, and data export. Data processing, statistical analysis, and data visualization were performed using R (Version 4.2.1) and RStudio (version 2022.02.01 461).

## 4. Results

### 4.1. General Patient Demographics

Participants had an age range from 51 to 100 years. The mean age of the participants was 76.35 years with a std. deviation of +/−9.25. The male/female ratio was 56.9%/43.1%. Table 3.

### 4.2. Univariate Analysis

There were no statistically significant differences between both groups in the univariate statistics. However, the group of acylcarnitines showed a tendency toward a lower concentration in cases with later recurrence (overall 65%). Furthermore, 10 of the 22 analyzed acylcarnitines (marked in light grey) showed a *p*-value <0.05, although no overall significance could be proven (Table 4).

### 4.3. Multivariate Analysis

Multivariate analyses were conducted to obtain an overview of the data set. Different possible confounding factors were evaluated to check whether they make a difference between both sample groups. However, neither age nor gender or the implantation of a drain appeared to influence the overall distribution of metabolites (Figure 3, Figure 4 and Figure 5).

### 4.4. Metabolism Indicators

The Met IDQ MetaboINDICATOR tool provided a set of preconfigured sums and ratios that proved to be particularly informative regarding certain clinical conditions or pathophysiological events. Metabolism indicators were calculated from cleaned and imputed data. If one of the metabolites necessary for the calculation of a given metabolism indicator was removed during the data-cleaning process, the metabolism indicator was not calculated. In this study, 62 out of 121 metabolism indicators could be calculated. The metabolites and metabolism indicators with significant concentration differences (*p* < 0.05) for the respective group comparison were compared between both groups (recurrence vs. no recurrence). MCAD and VLCAD, both involved in the process of fatty acid oxidation, showed noticeable drops in concentration. Only VLCAD showed statistical significance, although MCAD indicated a trend toward significance as well (Table 5).

## 5. Discussion

The major interest of the current work was the comparison of samples between different patient groups. Some of the patients were healed after a single surgery, while others presented with recurrent disease at a later stage and needed an operative revision. As cSDH is, in part, an inflammatory disease, it is of interest whether its recurrence is related to a higher degree of inflammation in the specific patient. Finding signs of metabolic inflammation was, thus, the goal of this analysis. The working hypothesis was that in patients with a later recurrence, more signs of metabolic inflammation might have been present. It was statistically ruled out (*p* = 0.18) that there was no difference between both groups regarding the pattern of drain addition. While disease recurrence probably cannot be entirely attributed to inflammatory processes, finding differences may offer further insights into the pathophysiology and prognosis of the disease.

The current study opted for a kit that enabled the analysis of many different metabolites at the same time through liquid chromatography–mass spectrometry. Among these were ceramides, sphingomyelin, and phosphatidylcholines, which were found in the cell membranes and play different roles in signaling pathways in the cell. The concentrations of amino acids, biogenic amines, and acylcarnitines were included in the kit as well. As there are no similar analyses conducted thus far, the absolute value of the data is inconclusive. Concentrations were of less interest to our study. They may, of course, be of comparative value to further studies in this field as a means of reference.

Our results did not offer many differences across the panel. Only a single substance class, that of acylcarnitines, showed differences between the two populations. All of the 22 different acylcarnitines that we analyzed showed a lower concentration in the cases that later presented with recurrence. For 10 of those 22 metabolites, the *p*-value was lower than 0.05, thus suggesting statistical significance. Even if the results may have failed to reach statistical significance overall, there was a clear trend toward a lower concentration of acylcarnitine (65% across the whole group) in the patients with recurrence.

The interpretation of these findings is difficult. There are no comparable data available, so a comparison is not possible. On the other hand, based on the knowledge of energy metabolism in inflammation, the results could potentially point in a similar direction. As cSDH is, in part, an inflammatory condition, a part of the energy could be produced through the citric acid cycle. The lower concentration of acylcarnitine could, however, mean that the transport of the broken-down fatty acids into the mitochondria is impaired or at least processed at a slower rate. This, on the other hand, could potentially mean lower energy resources because the compensation of tissue hypoxia through the oxidation of fatty acids is diminished. Inflammatory processes could, thus, be intensified, being the reason why these patients presented with higher recurrence rates. While these considerations are theoretical so far, they do seem plausible and may offer new research possibilities in the field.

What further shifted the focus toward the impaired energy metabolism through ß-oxidation was the fact that the kit detected a deficiency of MCAD and VLCAD for the group of cSDH samples with later recurrence. While MCAD merely showed a trend toward significance (*p* = 0.079), VLCAD deficiency seemed to reach statistical significance (*p* = 0.01). Both VLCAD and MCAD belong to the group of Acyl-CoA dehydrogenases and are mitochondrial enzymes. After acylcarnitines enter the mitochondria, they are separated so that Acyl-CoA can enter the ß-oxidation cycle. If, however, there is a deficiency of Acyl-CoA dehydrogenases, the ATP production does not function at normal rates. A deficiency of such dehydrogenases is known as part of genetic mutations. So far, there are no data suggesting that it could develop as a result of systemic conditions. The opposite seems to be the case, as there are data suggesting that Acyl-CoA deficiency might lead to an upregulation of inflammatory mediators [22]. Based on the very limited data in the literature, it is difficult to analyze the findings of our analyses. MCAD, for example, is a rare condition [23], and it seems highly unlikely that the patients who suffered hematoma recurrence in our series all had this syndrome. What the findings do show is that in those cases with later recurrence, the process of fatty acid oxidation might be compromised in some way. This, on the other hand, could, in turn, lead to an aggravation of the inflammatory condition, possibly leading to higher recurrence.

Although the results are not verified, they are generally consistent with theories that consider the molecular biological approach as a promising mode of treatment to avoid surgery, which has its own share of complications, especially because of the age of the patients with cSDH [24].

These findings suggest that understanding the role of metabolites in inflammation may further improve the knowledge of this complex process. Chronic subdural hematoma has proven to have a substantial inflammatory nature in recent years, so metabolite identification may further help determine the complex pathophysiological mechanism of its development.

In recent years, metabolomics has emerged as a promising field for further research on inflammation. This has been the case for systemic diseases, such as rheumatoid arthritis, as well as more localized diseases, such as Morbus Crohn’s or multiple sclerosis. Systemic inflammation leads to changes in the concentration of many metabolites in the body. Apart from the specific changes that can be observed in separate diseases, a common observation seems to present in the form of increasing energy requirements coupled with decreasing oxygen supply [25]. There is increasing evidence that the inflammatory environment is hypoxic [26]. Chronic hypoxia seems to upregulate the mechanism of fatty acid oxidation [27]. This process, known as ß-oxidation, breaks down fatty acids with the help of carnitine and enables the production of ATP via the citric acid cycle. During the process, acylcarnitines form as a means of transporting the fatty acids into the mitochondria. Acylcarnitines have also been identified as metabolic markers for errors in fatty acid oxidation and potentially play an important role in inflammation [28]. They have the potential to activate inflammation by activating proinflammatory signaling pathways, but the specific molecular and tissue target(s) remain to be identified [29].

Further studies on immunometabolism and its involvement in cSDH are necessary in order to develop a better conservative treatment of cSDH and its recurrence.

## 6. Potential Drawbacks

Recent data suggest that altered adipokine levels in patients with metabolic-associated diseases may be a part of inflammatory processes in the body [30,31,32]. Concomitant metabolic diseases, including diabetes and hypercholesterolemia, as well as cardiac and vascular diseases associated with metabolic diseases leading to coronary vessel occlusion and coronary syndrome, were also recorded in our patient population. Expectedly, a large number of the initial participants (69%) were perceived to be at high metabolic risk with multiple metabolic-associated diseases. Thus, a minor uncertainty remains as to whether the recorded metabolic changes in the hematoma fluid might not be at least in part attributed to these diseases. However, in the initial study, the possible relationship between cSDH recurrence and metabolic diseases was investigated, and no significant correlation could be found (*p* = 0.14). Therefore, the fact that a certain metabolic profile was found in patients who, at a later stage, presented with a recurrence does not seem likely to be caused by the metabolic conditions.

This study was also potentially limited by the relatively small initial sample size. Furthermore, some samples were lost in the process of analysis due to some deficiency of the required characteristics.

## 7. Conclusions

The current study presented a new approach to the research of cSDH, which is nowadays widely regarded to be at least partly inflammatory in nature. Metabolomics is, overall, a relatively new field for the research of inflammation. The measurements that we performed using mass spectrometry allowed us to gather data about the hematoma samples that have never before been studied. Therefore, it is difficult to interpret the information, and our conclusions should be considered only speculative. The results do, however, point in the direction of impaired fatty acid oxidations for cases with recurrence. As fatty acid oxidation plays an important role in inflammatory energy metabolism, the results suggest that inflammatory processes could be aggravated in cases with recurrence.

While our possible findings are neither proven through further analyses nor offer an obvious therapy option, their implications would not change everyday practice in the management of cSDH. They do, however, present a further possibility of research that might, in the future, be relevant to the therapy. Other possible fields of research might include the analysis of the hematoma fluid for the presence of adipokines in order to investigate a possible connection between cSDH and metabolic processes. Also, the investigation of the dura in affected patients for the presence of inflammatory mediator-receptors could provide more data on the inflammatory mechanisms involved in the pathophysiology of the disease.

## Figures and Tables

**Figure 1 diagnostics-13-03345-f001:**
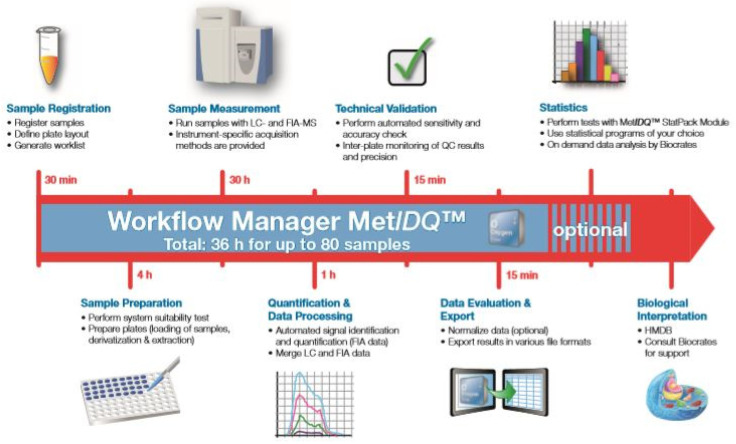
Workflow of the liquid chromatography–mass spectrometry.

**Figure 2 diagnostics-13-03345-f002:**
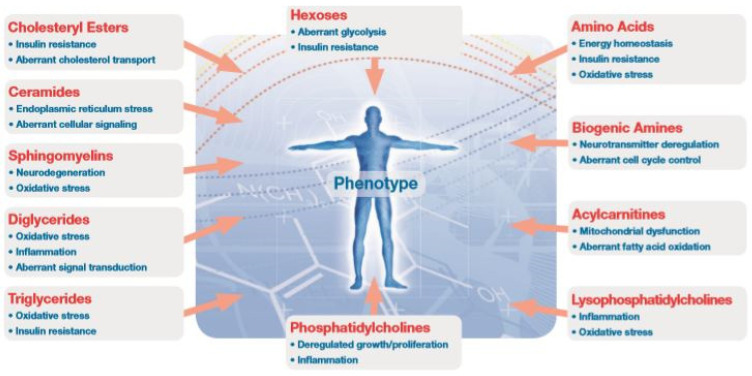
Metabolites analyzed.

**Figure 3 diagnostics-13-03345-f003:**
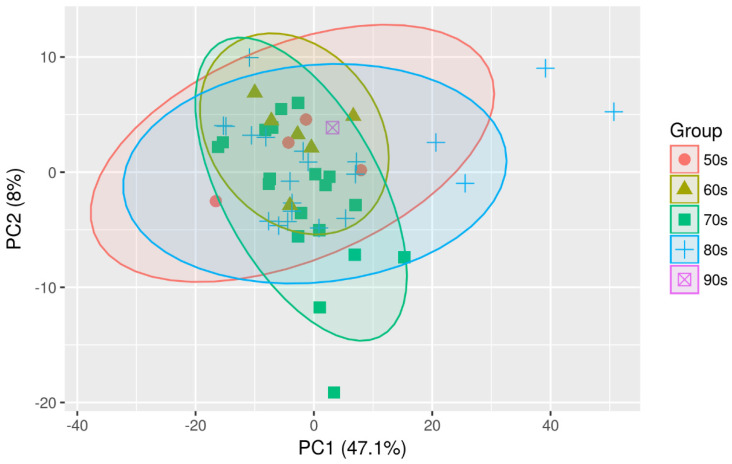
PCA-Scaling: Plot PC1 vs. PC 2-Age.

**Figure 4 diagnostics-13-03345-f004:**
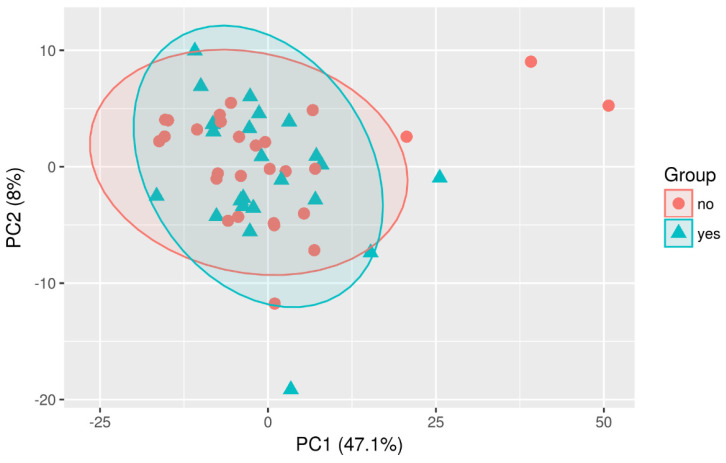
PCA-Scaling: Plot PC1 vs. PC2-Group comparison (recurrence vs. no recurrence).

**Figure 5 diagnostics-13-03345-f005:**
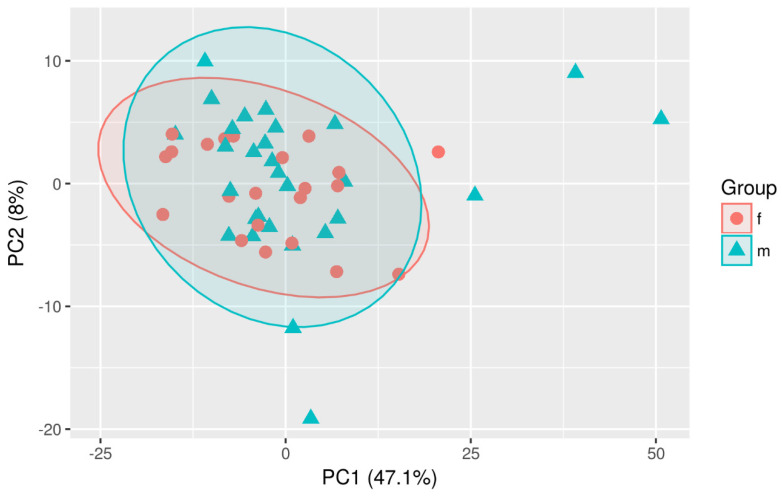
PCA-Scaling: Plot-PC1 vs. PC2-Sex.

**Table 1 diagnostics-13-03345-t001:** Inclusion/exclusion criteria for patient enrollment.

Inclusion Criteria	Exclusion Criteria
Signed written consent	Previous ipsilateral or infratentorial surgical procedure within the previous six months
Age > 18	Age < 18
Burr-hole craniostomy	

**Table 2 diagnostics-13-03345-t002:** Sample distribution.

	Total Samples*N* = 59	
Missing metabolite*N* = 8	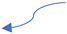	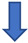	
	Complete metabolite profile*N* = 51
cSDH without recurrence*N* = 39	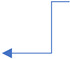	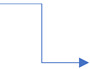	cSDH with recurrence*N* = 12

**Table 3 diagnostics-13-03345-t003:** Patients’ sociodemographic data.

	Sex						Recurrence of cSDH			
	Male		Female		Total		Yes		No	
Age	Freq	%	Freq	%	Freq	%	Freq	%	Freq	%
51–60	3	5.9	2	3.9	5	9.8	1	2	4	7.8
61–70	5	9.8	0	0	5	9.8	2	3.9	3	5.9
71–80	12	23.5	10	19.6	22	43.1	5	9.8	17	33.3
81–90	9	17.7	9	17.7	18	35.3	4	7.8	14	27.5
91–100	0	0	1	2	1	2	0	0	1	2
Total	29	56.9	22	43.1	51	100	12	23.5	39	76.5

**Table 4 diagnostics-13-03345-t004:** Acylcarnitine concentration.

Acylcarnitines	Ratio (Rec./No Rec.)	*p*-Value
AC(0:0)	0.83	0.103
AC(2:0)	0.82	0.134
AC(3:0)	0.83	0.238
AC(4:0)	0.77	0.124
AC(4:0-OH)	0.75	0.286
AC(5:0)	0.80	0.332
AC(6:0)	0.53	0.001
AC(8:0)	0.58	0.018
AC(8:1)	0.69	0.106
AC(10:0)	0.58	0.012
AC(10:1)	0.69	0.081
AC(12:0)	0.47	0.010
AC(12:1)	0.56	0.007
AC(14:0)	0.42	0.006
AC(14:1)	0.40	0.002
AC(14:2)	0.51	0.017
AC(16:0)	0.76	0.323
AC(16:1)	0.50	0.012
AC(16:2)	0.56	0.049
AC(18:0)	0.70	0.234
AC(18:1)	0.75	0.259
AC(18:2)	0.86	0.599

**Table 5 diagnostics-13-03345-t005:** Metabolism indicators.

Metabolism Indicator	Ratio	*p*-Value
MCAD Deficiency (NBS)	0.70	0.079
VLCAD Deficiency (NBS)	0.53	0.01

## Data Availability

Data will be provided by the corresponding author on request.

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
