# Peer review of "Immunometabolic Profiling of Chronic Subdural Hematoma through Untargeted Mass Spectrometry Analysis: Preliminary Findings of a Novel Approach"

_diagnostics, 2023, doi:10.3390/diagnostics13213345_

Round 1

Reviewer 1 Report

The authors present an innovative approach to cSDH research with interesting and promising initial results, that they interpret with adequate caution.

I recommend to accept the work for publications after only minor revision.

In the introduction, it should be acknowledged that the described pathophysiological connections leading to cSDH and cSDH recurrence still represent theories up for discussion.

Besides intrinsic risk factors, there are some important external risk factors for cSDH recurrence that may be relevant in the current dataset. Can the authors determine whether (repeat) head trauma occured in the three months after the surgery? And more importantely, did any patients receive anticoagulants and / or antiplatelets or NSAR for pain medication?

The authors might consider removing Figure 2, as I feel it does not constitute a relevant addition, nor does it seem necessary to elucidate descritptions in the manuscript.

Author Response

Dear reviewer,

thank you very much for your valuable comments. We tried to implement them and add the missing information. 

Regarding your first comment, you are of course right, that the current understanding is also another theory so we may have to change it again over time.

To our knowledge, no further head trauma had occured during the the follow-up period. In terms of medication, no antiplatelet drugs or anticoagulants were administered. NSARs however are part of the routine pain therapy, so many of the patients received it, as long as there were no allergies.

As your last remark you recommend the removal of Figure 2. We thought that it might be of interest to readers to see which metabolites exactly were measured. If you nevertheless feel that the addition of the figure is unnecessary we would gladly remove it from the manuscript.

Sincerely,

Y. Bozhkov

Reviewer 2 Report

Dear colleagues. Your've got collected an interesting facts. But they can't be an adecvate indicators to the practical diagnostics. And we have no potential tools which can arise in a nearest future. It's better to submit this paper to the journals more relative to the general pathology. 

The language level is very professional. There was no such thing that the thread and logic of presentation were lost. 

Author Response

Dear reviewer,

thank you for your valuable remarks. We have to agree with you, that the current results can not be implemented into everyday routine in the current state. I hope we did not leave the impression that we are discovering some form of important diagnostic tool for cSDH. The way we iterpret our work is as follows: The current understanding of cSDH pathophysiology is also just a theory like many before. Through the use of metabolomics we try to uncover a further piece of the puzzle and check whether the findings are in line with what is currently hypothetised. If this theory in time proves to be correct and cSDH and its recurrence are really mostly inflammatory in nature, may be we can implement some form of therapeutic regime (similar to current use of dexamethasone). This might reduce the need for surgery or even make it obsolete as a whole, which of course would be of benefit to patients. 

We are very humble about our findings and know we are not revolutionizing the diagnostics of cSDH but still we believe they they are of interest to further research. As Diagnostics is running the special issues on cSDH we felt that it would be the right platform for such findings. 

Sincerely,

Y. Bozhkov

Reviewer 3 Report

The authors present the results of a clinical study that is part of a prospective randomized trial concerning surgery of chronic subdural hematoma (cSDH) with adjuvant use of periosteal drains, which aimed to compare the inflammatory profiles of cSDH fluids between patients with and without later disease recurrence focusing on metabolites.  The authors used liquid chromatography–mass spectrometry to examine the selected probes. The results suggest that inflammatory processes could be aggravated in cases with later recurrence. For a better presentation of the data, the following issues need to be clarified:  

        1. The study was potentially limited due to the small size of the sample. The title should clearly mention “preliminary findings”. 
        2. Please describe all acronyms used (cSDH) in the first mention (Abstract)  
        3. It is recommended that the authors begin the Discussion by assessing the most relevant data of their study. 
        4. A brief  concluding comment on other possible lines of future research on the topic presented would be appreciated.   

Author Response

Dear reviewer,

thank you for your valuable remarks. 

According to your recommendation we revised the manuscript title, because the relatively small sample size is indeed of importance.

We decided to reorder the introduction and discussion parts as you suggested, so that the text is more structured. This way, it focusses on the main topic of our work better. The changes have been highlighted in grey but they only include reorganisations and no new information.

Finally, further research possibilities were described in the conclusion part. As of now, we are even working on the dura analysis, which we hope to be able to present in the near future.

Sincerely,

Y. Bozhkov

Round 2

Reviewer 2 Report

Dear colleagues, I would like to wish you good luck in further studying dysmetabolic processes in patients with chronic subdural hematomas.